# Individualized Prediction of SARS-CoV-2 Infection in Mexico City Municipality during the First Six Waves of the Pandemic

**DOI:** 10.3390/healthcare12070764

**Published:** 2024-03-31

**Authors:** Mariel Victorino-Aguilar, Abel Lerma, Humberto Badillo-Alonso, Víctor Manuel Ramos-Lojero, Luis Israel Ledesma-Amaya, Silvia Ruiz-Velasco Acosta, Claudia Lerma

**Affiliations:** 1Master’s Program in Biomedical Sciences, Institute of Health Sciences, Autonomous University of the State of Hidalgo, San Agustín Tlaxiaca 42160, Mexico; vi244818@uaeh.edu.mx; 2Area of Psychology, Institute of Health Sciences, Autonomous University of the State of Hidalgo, San Agustín Tlaxiaca 42160, Mexico; luis_ledesma@uaeh.edu.mx; 3Jalalpa el Grande Health Center, Mexico City Health Secretariat, Mexico City 01377, Mexico; humbertobadilloalonso@gmail.com; 4Health Jurisdiction Alvaro Obregon, Mexico City Secretary of Health, Mexico City 01470, Mexico; victorlojero1@gmail.com; 5Instituto de Investigaciones en Matemáticas Aplicadas y en Sistemas (IIMAS), Universidad Nacional Autónoma de México, Mexico City 04510, Mexico; silvia@sigma.iimas.unam.mx; 6Centro de Investigación en Ciencias de la Salud (CICSA), FCS, Universidad Anáhuac México Campus Norte, Huixquilucan Edo. de Mexico 52786, Mexico; 7Department of Molecular Biology, Instituto Nacional de Cardiología Ignacio Chávez, Mexico City 04480, Mexico

**Keywords:** COVID-19, statistical modeling, logistic regression, risk factors, symptoms

## Abstract

After COVID-19 emerged, alternative methods to laboratory tests for the individualized prediction of SARS-CoV-2 were developed in several world regions. The objective of this investigation was to develop models for the individualized prediction of SARS-CoV-2 infection in a large municipality of Mexico. The study included data from 36,949 patients with suspected SARS-CoV-2 infection who received a diagnostic tested at health centers of the Alvaro Obregon Jurisdiction in Mexico City registered in the Epidemiological Surveillance System for Viral Respiratory Diseases (SISVER-SINAVE). The variables that were different between a positive test and a negative test were used to generate multivariate binary logistic regression models. There was a large variation in the prediction variables for the models of different pandemic waves. The models obtained an overall accuracy of 73% (63–82%), sensitivity of 52% (18–71%), and specificity of 84% (71–92%). In conclusion, the individualized prediction models of a positive COVID-19 test based on SISVER-SINAVE data had good performance. The large variation in the prediction variables for the models of different pandemic waves highlights the continuous change in the factors that influence the spread of COVID-19. These prediction models could be applied in early case identification strategies, especially in vulnerable populations.

## 1. Introduction

SARS-CoV-2 is a type of coronavirus that causes the COVID-19 disease, which has several symptoms. In Mexico, during the first months of the pandemic, about 24% of patients needed hospitalization [1,2]. Since the beginning of the COVID-19 pandemic, the world has been overwhelmed by the number of infections in very short periods [2]. Over the last two years, the pandemic has developed through different infection peaks with different SARS-CoV-2 virus variants. Up until the beginning of 2023, six waves of contagion had been reported globally [2]. Alpha and beta are known to affect humans the most out of these strains. In Mexico, the waves corresponded to those reported worldwide by the WHO, reporting up to 29 November 2022, 7.13 million infections and 330,000 deaths, with almost 2500 new infections confirmed daily [1,2,3,4].

At the beginning of the pandemic, in Mexico, the only diagnostic methods were SARS-CoV-2 PCR tests from the Diagnostic and Epidemiological Reference Institute (InDRE), which had 98% efficacy. The diagnostic methods were scarce compared to the number of daily infections [1]. Initially, in Mexico City, tests were only available to patients who visited specialized centers designated for treating the disease and displayed apparent signs of severe symptoms. Later, private laboratories began conducting tests, but at a high cost of up to USD 200 to USD 300. In Mexico, testing was only available in large hospitals and COVID centers in cities, and rural populations had virtually no access to them or treatment for the disease; many people still cannot afford to pay for them or must travel long hours and wait long periods to obtain one from the public health system [1,2,5,6,7,8].

With the emergence and spread of the virus in 2020, several proposals have been published to solve the problem of the lack of testing that led to many more infections. Statistical models were developed by different methods, from mathematical analysis to artificial intelligence [9,10,11,12,13,14,15,16]. The new methods, mainly using similar variables in their prediction models, use known data from patients diagnosed as positive for the virus by laboratory tests. Known data may include symptomatology, comorbid diseases, demographic characteristics, and recent mobility, although some models included data from other laboratory tests obtained from hospitalized patients [9,10,11,12,13,14,15,16]. Furthermore, new methods have been developed to diagnose COVID-19 by applying artificial intelligence (AI) (up to 99% accuracy), “immunoassay” and “biosensor-based identification” (66–95% accuracy), mobile devices or smart rings (up to 99% predicted), “plasma biosensors”, and “electromechanical biosensors” as well as the analysis of chest X-rays with artificial intelligence (98% of hits), the use of cough and breath sounds, and “COVID-19 symptom tracker” (98% sensitivity) [16].

In Mexico, diagnostic test data have been collected by the National Epidemiological Surveillance System (SINAVE). The variables in the SINAVE databases include patient data, the symptoms of the virus, contact with a suspicious case, contact with animals, disease evolution, comorbid diseases, and information from the sampling and test results [3]. Although epidemiological models have been developed for the prediction of COVID-19 tests or to estimate the risk of hospitalization or mortality, no models have been reported for the individualized prediction of SARS-CoV-2 infection for the Mexican population [9,10,11,12,13,14,15,16].

The objective of this work was to develop a model for the individual test prediction of COVID-19 based on SINAVE data for patients with suspected SARS-CoV-2 infection who underwent a test at the Alvaro Obregon Jurisdiction health centers in Mexico City during the six waves of contagion that occurred between 2020 and 2023.

## 2. Materials and Methods

### 2.1. Study Design and Participants

The study design was observational and retrospective. The study was conducted with data from patients who attended the health centers of the Alvaro Obregon Jurisdiction of Mexico City between April 2020 and January 2023. All patients were selected within 3 weeks at the peak of each of the waves according to data reported in SINAVE 3 (Table 1). We included patients who requested the PCR test due to suspicion of COVID-19 and who were older than 18 years. Patients with inconclusive results in their PCR test were excluded. The research protocol was approved by the Ethics and Research Committee of the Universidad Autónoma del Estado de Hidalgo (protocol number Comiteei.icsa 017/2022 approved on 18 March 2022). All procedures adhered to the principles stated in the Declaration of Helsinki.

### 2.2. Procedure for Obtaining Study Data and Variables

Data were obtained from the Epidemiological Surveillance System for Viral Respiratory Diseases (SISVER/SINAVE) database of the Ministry of Health of Mexico, which contains the information variables of those patients who came to the health centers of the Alvaro Obregon Jurisdiction for suspicion of infection with the COVID-19 virus [3]. These data were collected by the health professionals attending to the patients and recorded in the system with an identifier number per patient.

The samples of each of the patients were taken at COVID centers especially designated for this type of care using pharyngeal and nasopharyngeal exudates; these were sent to InDRE, and the results were subsequently recorded in the SINAVE database. The results were recorded as a field in the database named “Final result” and were classified as “SARS-CoV-2”, “Negative”, “Rejected”, “Unsuitable”, “Non-subscribed”, “Positive”, and empty fields. The results classified as “SARS-CoV-2” and “Positive” were considered positive tests and recoded as 1, while the results classified as “Negative” were considered negative tests and recoded as 0 for further analysis. The results classified with other criteria (“Rejected”, “Unsuitable”, Non-subscribed”, or empty fields) were removed as they were not suitable for analysis [3].

The variables included in the regression analysis were selected from a literature review regarding the main characteristics presented by patients diagnosed as positive within each wave as it was shown that some of these characteristics changed concerning the SARS-CoV-2 virus subtypes; these were included in a general way for the models since the database did not record information on the variant of the virus with which the positive patient was infected [3,9,10,11,12,13,14,15,16,17,18,19].

The following variables were included: sex, age, main comorbidities for the Mexican population (diabetes, hypertension, and obesity [9]), smoker, contact with a suspicious case, and presence of symptoms (fever, cough, odynophagia, dyspnea, irritability, diarrhea, chest pain, chills, headache, myalgia, arthralgia, rhinorrhea, polypnea, vomit, abdominal pain, conjunctivitis, cyanosis, anosmia, dysgeusia). All variables except age and sex were classified as “YES” and “NO” and were recorded as 1 and 0, respectively. Cases with captured data such as “IGNORED” and empty fields were excluded [3].

The SISVER/SINAVE database was consulted to obtain all of the cases registered for the health centers in the Alvaro Obregon Jurisdiction during each peak of the first six waves of the pandemic (Table 1). The dates with the highest number of cases of each wave were identified. For waves 2, 3, 4, 5, and 6, an interval of one week before and one week after the week with more cases was selected (total sampling period of 3 weeks per wave). During wave 1, there was little availability of evidence, and the criteria for defining a suspicious case were stricter, resulting in a low rate of negative cases. Therefore, for the wave 1 sample, an interval of two weeks before and two weeks after the week with more cases was selected (a total sampling period of 5 weeks).

### 2.3. Statistical Analysis

Since a Kolmogorov–Smirnov test showed that age had no normal distribution (*p* < 0.01), it was described as median (25th percentile–75th percentile) and compared between waves and between positive or negative results of each wave by Mann–Whitney U tests. The nominal variables were described as absolute values and percentages and compared between waves and between test results (positive or negative) with the Chi-square test.

To predict the SARS-CoV-2 test result, binary logistic regression analyses were performed, obtaining a regression model for each of the six waves. Only the variables with significant differences between each wave’s positive and negative groups were included. We used the conditional forward method of binary logistic regression with the goodness of Hosmer–Lemeshow adjustment. An analysis of influential points was performed using Cook’s distance for the final model of each wave. The odds ratio (OR) with a 95% confidence interval (CI) is reported for the significant variables in each model. The statistical analysis was performed with SPSS version 21.0. A value of *p* < 0.01 was considered significant.

## 3. Results

Table 2 shows the characteristics of the study participants. The asterisk symbols (*) show that wave 1 had a greater age, a higher percentage of women, and a larger proportion of smokers, hypertension, obesity, diabetes, and contact with a suspicious case compared to subsequent waves (2 to 6). Compared to wave 2 (symbol &), the age in subsequent waves 3 and 4 was lower, and was higher in waves 5 and 6. Wave 2 also had a higher proportion of smokers (compared to waves 3, 5, and 6), a higher proportion of hypertension and diabetes (compared to waves 3 and 4), a lower proportion of obesity (compared to wave 4), and less contact with a suspicious case (in the subsequent 3 waves 3, 4, 5, and 6). Wave 4 had the lowest proportion of comorbidities (hypertension, obesity, diabetes). Compared to wave 3 (symbol #), waves 4 and 5 had significant differences in age, number of smokers, the proportion of patients with hypertension, obesity (only in wave 4), diabetes (only in wave 5), and contact with a suspicious case (only in wave 5). Wave 6 (symbol ≠) showed statistically significant differences with the other waves in most variables.

Table 3 shows the sociodemographic and morbidity data of the participants compared with the test results (positive or negative); the asterisks indicate statistically significant differences between positive and negative within the same wave (*p* < 0.01). In wave 1, the different variables between positive and negative (indicated with *) were: age, sex, hypertension, diabetes, and contact with a suspicious case. In wave 2, the different variables between positive and negative are age, smoking, hypertension, obesity, diabetes, and contact with a suspicious case. In wave 3, the positive ones differed from the negative ones in age, sex, smokers, obesity, diabetes, and contact with a suspicious case. In wave 4, the positive cases differed from the negative ones in hypertension, obesity, and contact with a suspicious case. In wave 5, the variables with the differences between positive and negative were age, smoking, hypertension, obesity, diabetes, and contact with a suspicious case. Wave 6 showed differences between age, diabetes mellitus, and contact with a suspicious case.

Figure 1 and Appendix A compare the prevalence of symptoms in patients with positive vs. negative tests within each wave. The symptoms were ordered from the most prevalent to the least prevalent within the following groups of symptoms: respiratory system (cough, rhinorrhea, dyspnea, cyanosis, and odynophagia), digestive or gastrointestinal system (anosmia, dysgeusia, diarrhea, abdominal pain, and vomit), pain related (headache, myalgia, arthralgia, and chest pain), and other types (fever, chills, irritability, and conjunctivitis). In wave 1, the percentage of symptoms among the positive and negative patients was different in most variables: cough, rhinorrhea, dyspnea, polypnea, anosmia, dysgeusia, headache, myalgia, arthralgia, chest pain, fever, and chills. In waves 2 and 3, all variables presented statistically significant differences between positive and negative, which were cough, rhinorrhea, dyspnea, polypnea, cyanosis, odynophagia, anosmia, dysgeusia, diarrhea, abdominal pain, vomiting, headache, myalgia, arthralgia, chest pain, fever, chills, irritability, and conjunctivitis. Wave 4 presented differences in the following variables: cough, rhinorrhea, odynophagia, diarrhea, abdominal pain, headache, myalgia, arthralgia, chest pain, fever, chills, and irritability. Wave 5 presented differences in cough, rhinorrhea, dyspnea, polypnea, odynophagia, dysgeusia, diarrhea, abdominal pain, headache, myalgia, arthralgia, chest pain, fever, chills, irritability, and conjunctivitis. In wave 6, only cough, rhinorrhea, dyspnea, and vomit showed a statistically significant prevalence between patients with positive and negative tests.

Figure 2 presents the binary logistic regression analysis results with models to predict the outcome of the SARS-CoV-2 test (positive or negative) as a dependent variable. A different model was calculated in each wave including independent variables with significant differences between positive and negative (Appendix A). For example, the initial wave 1 model included age, female sex, hypertension, diabetes, contact with a suspicious case, cough, rhinorrhea, dyspnea, polypnea, anosmia, dysgeusia, headache, myalgia, arthritis, chest pain, fever, and chills. The variables not initially included in the wave 1 model, indicated as “Not included,” were smoker, obesity, cyanosis, odynophagia, diarrhea, abdominal pain, vomiting, irritability, and conjunctivitis. When applying the logistic regression analysis in model 1 with the later conditional method, the variables that did not have a significant contribution to the dependent variable, indicated as “Eliminated”, were eliminated: age, hypertension, diabetes mellitus, contact with a suspicious case, rhinorrhea, polypnea, anosmia, myalgia, arthralgia, chest pain, and chills. The final wave 1 model had the following variables with a significant odds ratio (*p* < 0.05): female sex, diabetes, cough, dyspnea, dysgeusia, headache, and fever.

Applying the same logistic regression analysis strategy, the final models of subsequent waves had different predictor variables (Figure 2). The final wave 2 model had age, smoking, obesity, contact with a suspicious case, cough, rhinorrhea, dyspnea, polypnea, odynophagia, anosmia, dysgeusia, vomiting, headache, arthralgia, chest pain, fever, chills, irritability, and conjunctivitis as predictors, while the final wave 3 model had age, female sex, cough, rhinorrhea, cyanosis, odynophagia, anosmia, dysgeusia, diarrhea, abdominal pain, headache, myalgia, arthralgia, fever, chills, and irritability as predictors. The final wave 4 model had cough, odynophagia, myalgia, fever, chills, and irritability as the final predictor variables. The final wave 5 model had age, cough, rhinorrhea, headache, fever, and chills as predictors. Predictors included in the final model for wave 6 included age, contact with a suspicious case, cough, dyspnea, and vomit. For comparative analysis, we verified whether the confidence intervals for the odds ratio in the different waves overlapped (Appendix A). The odds ratio of age during wave 5 was larger than during wave 3. The odds ratio of being a smoker was lower during wave 3 than during wave 2. Compared to waves 1 and 2, the odds ratio of cough was consistently higher during waves 3, 4, and 5. Odynophagia also had a higher odds ratio during waves 3 and 4 than during wave 2. Headache had higher odds ratios during waves 2, 3, and 5 than during wave 1. The odds ratio of fever during wave 3 was larger than in wave 2, and wave 5 also had a larger odds ratio of fever than wave 2 and wave 4. Finally, the odds of irritability were larger in wave 4 compared to waves 2 and 3.

Table 4 shows the performance results of the models. The models’ determination coefficients (R^2^) ranged from 0.051 to 0.535. The global predictions also varied according to the wave, with an average overall accuracy of 73% (range between 63% and 82%), an average sensitivity of 52% (range between 18 and 71%), average specificity of 84% (range between 71 and 92%), average positive predictive value of 68% (range between 55 and 76%), and an average negative predictive value of 73% (range between 56 and 76%). In the final models, the Cook statistic for influence was calculated, and we did not find any influential observations (Appendix A).

## 4. Discussion

### 4.1. Main Contribution

The main contribution of this work is to demonstrate the development of statistical models for the individual prediction of SARS-CoV-2 virus infection from sociodemographic data and symptoms usually collected in Mexico’s epidemiological surveillance system. As a test population, the work focused on a sample by the census of all cases registered in the Alvaro Obregon Jurisdiction, one of the most populated with great socioeconomic heterogeneity, in one of the most populous cities in the world, Mexico City. This municipality has remained one of the most affected since the beginning of the pandemic and has even been the epicenter of cases of COVID-19 infection in Mexico. The prediction models showed very good performance, so they are a potential new tool to deal with new waves of contagion. They promote an early risk diagnosis and thus favor measures that reduce infections. The models also show that the prediction variables varied widely between the six waves, highlighting the continuous change in factors that influence the spread of COVID-19, hence the importance of considering statistical models for individual prediction that can be updated, as these factors evolve with data from the same population.

### 4.2. Comparison to Other Models for Personalized Prediction of SARS-CoV-2 Infection

Some similar models have already been developed in different parts of the world, mainly in the United States. There is a report based on variables and methods like the present work in California [9], with a model based on symptoms of smell and taste, which also included morbidities and the symptomatology most frequently reported until then with that variable in the year 2020, where they also applied logistic regression models with a correct prediction of 75%. Another previous work [10] also determined the probability of having the disease or not from surveys and risk factors. Another model developed in Alabama [14] used a risk calculator from an electronic recording of the most reported risk factors such as the mentioned symptoms, smokers, and psychological factors. On the other hand, in Minnesota, they developed another model that used the same variables plus patient mobility and applied multivariate logistic regression, which showed the predictor variables with the highest probability of being positive patients.

Most prediction models for COVID-19 tests are applied using logistic, binary, or multivariable regression [9,10,11,12,13,14,15,16], whose results generally obtain percentages of between 70 and 80% of correct prediction. However, other models are also applied to test predictions for such a disease. Some have applied artificial intelligence to predict the pandemic’s development or make individualized prediction models. One example was developed in Israel, in which they used the basic information and symptomatology of patients as dichotomous variables, plus contact with positive cases, achieving an accuracy reported as high [11]. Some of the latest models also apply electronic devices such as smartwatches or even new materials to measure the bio-signals that will be analyzed [16].

The current work focused on applying the most precise and straightforward statistical approach to unravel the patient characteristics and symptoms that would better predict an individual’s positive or negative SARS-CoV-2 test. The approach was applied to a valuable large dataset from a densely populated municipality in one of the world’s most inhabited urban areas. This is the first work that shows the variables that better predicted an individual’s COVID-19 infection during each of the first six waves of the pandemic. Our results are an incentive for further studies when there are more advanced methods to improve the accuracy, based, for instance, on artificial intelligence. Furthermore, structural equation modeling based on the groups of symptoms described in Figure 1 could help identify latent predictor variables from particular systems that are most affected in certain patients or by specific virus variants.

### 4.3. Heterogeneity in the Variables of Models for Individual Epidemic Waves

Different variables were used in the prediction model of each wave, which were selected from the significant differences between each wave’s positive and negative cases. As already mentioned, at each of the peaks of each wave, different variants of the SARS-CoV-2 were presented, so the variables in the models changed depending on these characteristics. In addition, after the first wave, vaccination schemes began in older people, and as the pandemic progressed, they continued to be vaccinated by age group to reach the youngest. This could influence changes in the behavior of the variables of the models as the pandemic progressed, being that the fifth wave raised the age again, with the hypothesis that this sector of over 50 years was the first to be vaccinated and by this time, adults over 60 were already receiving the vaccination scheme again [9,10]. Currently, COVID-19 testing has become more accessible to the population, both in the public and private sectors, which influenced an increase in negative tests in the databases [8]. The determinant variables in this study and the literature agree with those previously reported in other populations [1,2,3,17,19,20,21,22,23,24,25]. It should be noted that no models had been made with the six waves separately and in general, so comparing the present work with previous reports regarding individual waves was impossible.

It can be observed, for example, that certain variables were not included in the model from the significance, and that others were excluded during the regression analysis. Age was one of the variables included in waves 2, 3, 5, and 6, but not in 1 and 4, where there was either no vaccination or the first block became the one that had the longest with the scheme. Sex was the only variable predictor in wave 1, being a smoker only in wave 2. In contrast, comorbidities such as hypertension and diabetes did not appear in any model, obesity in wave 2, while contact with a suspicious case was a predictor only in waves 2 and 6.

Regarding symptomatology, cough was present in all waves as a predictor, dyspnea was included in waves 1, 2, and 6, polypnea in wave 2, cyanosis in wave 3, odynophagia in waves 2, 3, and 4, dysgeusia in 1, 2, and 3, and diarrhea and abdominal pain in wave 3. Vomit only was a predictor in wave 2 and headache occurred in all waves except in waves 4 and 6, myalgia in waves 3 and 4, arthralgia in waves 2 and 3, and chest pain in wave 2. Fever was present in all waves; chills were not a predictor in wave 1. Irritability was in waves 2, 3, and 4, and conjunctivitis only in waves 2 and 6. As can be seen, the gastrointestinal symptoms agreed with the appearance of Delta. Fever and cough were in all waves; most variables were present in at least three of the waves, and others were only excluded in one such as headache and chills. At the same time, it can be concluded that the first wave included the largest number of variables.

Despite the high heterogeneity of variables included in the final models for the six waves, some predictors in more than one wave showed changes in the magnitude of the odds ratio when compared between different waves (Appendix A). For instance, cough, the symptom predictor in all waves, had consistently larger odds ratios during waves 3, 4, and 5 compared to the initial waves 1 and 2. This consistent increase in the relative weight that a symptom (i.e., cough) had on the risk of a positive SARS-CoV-2 test could be related to differences in the symptoms experienced in most patients with different variants (Table 1). However, in most variables present in models of two or more waves, there was an overlap in the confidence intervals of the odds ratio, and therefore, observing a changing pattern in the relative weight of a symptom throughout the pandemic was most likely an exception rather than the rule.

### 4.4. Potential Practical Applications

The results of this work show that it is possible to create new methods for predicting COVID-19 tests by adapting them to the characteristics of the study population. In contrast, infections will increase with new waves in the future, and studies suggest that it will be a endemic disease. The virus continues to change, and today, more than ever, people must have access to alternatives to laboratory methods that are also faster. The epidemiological implications of this research work are the possible reduction in infections in the Mexican population by providing new tools to care for their health that are faster and cheaper. This also opens the door for the further development and investigation of such methods applied to the Mexican population. As mentioned, each population has its specific characteristics, and the variables considered for the prediction models change between populations and within the same population over time. Among the clinical considerations of the results, we can see the possible relief of the medical services that continue to provide services to COVID-19 patients, and although vaccination schemes have helped the population to face the pandemic, we cannot ignore the fact that the SARS-CoV-2 virus has come to stay as an endemic disease [8].

In many developing countries in Latin America, the minimum salary ranges from USD 3.61 in Venezuela to USD 687 in Costa Rica, with an average of USD 360.44 [26]. The cost of COVID PCR tests in these countries can range from 10–100% (USD 62) of this amount, making it difficult for many people to access them [27,28]. Additionally, people living in extreme poverty, particularly in Africa, often need more resources and technology for the early assessment of COVID-19. Thus, the number of infections is still being determined, particularly in low-income countries. This study proposes a COVID-19 detection model that could be an accessible and cost-effective alternative to PCR tests, which is most relevant for vulnerable populations with difficult access to clinical facilities (either public or private).

### 4.5. Study Limitations

Among the study’s limitations are the above-mentioned variables and the fact that only patients with severe symptoms were tested during the first wave. Hence, the test results had few negatives to include in the model. Another major area for improvement is that recent travel records were not included in the database since this variable was considered a potential predictor at the beginning of the study and according to the literature review. Another limitation of the study is the lack of information on the virus variant from which patients were infected since the test result was only classified as positive or negative. Therefore, it was not possible to classify the variables corresponding to each variant of the SARS-CoV-2 virus. However, the variable not included that is probably the most important is the vaccination scheme. Since vaccination began, many variables have changed simultaneously such as symptomatology, the number of patients who went to health centers to be tested, and the percentages of positive and negative patients. However, another major limitation of the study is that vaccination-related information was not considered as it was only recently included in the SINAVE database, and no data were available for most waves.

The variables selected for the initial model of each wave were identified by a comparison of positive versus negative tests from the entire dataset of each wave, and then using the forward conditional method to test the effect of each added variable on the model’s goodness of fit. Although this is a common (and widely accepted) approach to identifying the model variables that we considered good enough as a first approximation to obtain an individualized prediction of SARS-CoV-2 infection in Mexico, future studies are needed to address the potential model selection bias through more robust methods such as cross-validation or bootstrapping.

Future studies are also needed to identify sources of data variation and prediction results between waves. Finally, the SINAVE database includes information on the symptoms’ presence, but not the severity, which may help to improve the prediction accuracy. However, our study was primarily based on patients from outpatient clinics, where the severity of symptoms was mild or moderate and comprise the most significant proportion of infected patients in the overall population. Further studies are required to test the feasibility of using the presence of early symptoms during the acute infection to predict severe chronic symptoms such as fatigue, syncope, and delirium or brain fog, associated with the development of long-COVID syndrome. 

## 5. Conclusions

In the present work, individual test prediction models were developed in a sample of Mexico City that showed a good percentage of global correct prediction. The models developed here could be very useful for the health care of the Mexican population. Because these were based on a simple methodology and data publicly available through the country’s health system, these models are likely to achieve, with more sophisticated methods such as structural equation modeling or artificial intelligence, an increase in the sensitivity and specificity. Moreover, our findings show a high heterogeneity in the variables that comprised each model of the six waves, highlighting the dynamic change in the primary factors driving the epidemic. Therefore, we recommend that further research on prediction models of SARS-CoV-2 infection should consider from the start all the symptoms reported in the SINAVE database and other factors such as age, sex, and comorbidities. Our results encourage the future inclusion in the SINAVE database of information regarding the vaccination history and the severity of the symptoms. This may be relevant for models to predict the acute infection with SARS-CoV-2 (as in the present work) and for predicting long-term outcomes (such as long-term COVID syndrome) based on the early response during the acute infection.

## Figures and Tables

**Figure 1 healthcare-12-00764-f001:**
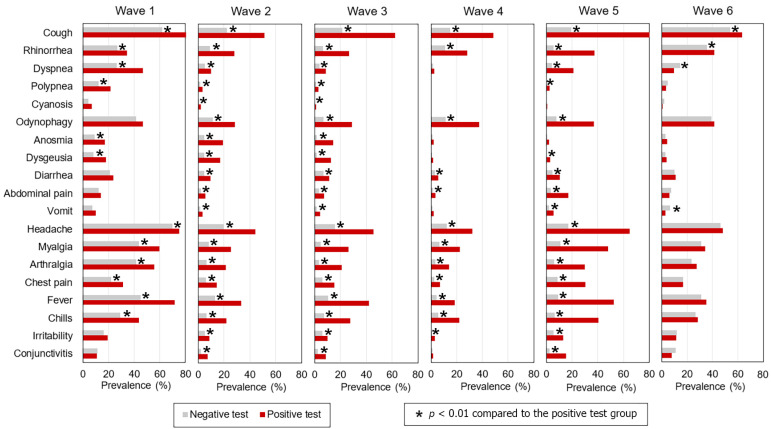
Prevalence of symptoms (positive and negative for each wave and general).

**Figure 2 healthcare-12-00764-f002:**
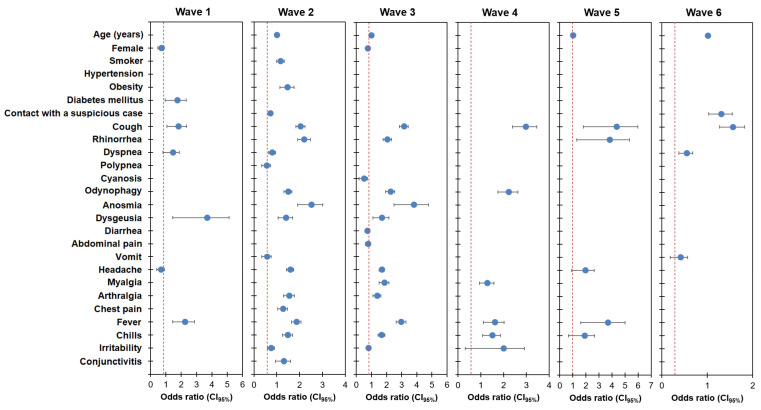
Binary logistic regression analysis to predict an individual SARS-CoV-2 test for each wave.

**Table 1 healthcare-12-00764-t001:** Characteristics of the data for each of the six waves of the COVID-19 pandemic.

	Wave 1	Wave 2	Wave 3	Wave 4	Wave 5	Wave 6
Period	19 April–21 June 2020	4–24 January 2021	19 July–8 August 2021	3–23 January 2022	20 June–10 July 2022	27 November 2022–14 January 2023
Main variant	Original strain	Alpha, Beta, and Gama	Delta	Omicron	Omicron (sub-lineages)	Omicron (sub-lineages)
Total patients	2081	13,535	15,368	6159	1177	3332
Excluded patients	72	1196	2072	822	167	840
Final sample size	2009	12,339	13,296	5337	1010	2492

**Table 2 healthcare-12-00764-t002:** Sociodemographic characteristics and comorbidities of the study population. Data are shown as absolute value and percentage or median (25th–75th percentile).

	Wave 1	Wave 2	Wave 3	Wave 4	Wave 5	Wave 6
**Variable**	(N = 2009)	(N = 12339)	(N = 13296)	(N = 5337)	(N = 1010)	(N = 2492)
Age (years)	46 (35–57)	40 * (29–52)	35 *^&^ (26–48)	38 *^&#^ (29–51)	43 *^&#$^ (31–58)	42 *^&#$^ (31–56)
Sex		*	*	*	*^&$^	*^&#$^
Female	1006 (50%)	6668 (54%)	7306 (55%)	2854 (53%)	590 (58%)	1491 (59.8%)
Male	1003 (50%)	5671 (46%)	5990 (45%)	2483 (47%)	420 (42%)	1001 (40.2%)
Smoker	240 (12%)	1234 (10%)	1036 (8%) *^&^	560 (11%) #	41 (4%) *^&#$^	152 (6%) *^&#$≠^
Hypertension	367 (18%)	1124 (9%) *	818 (6%) *^&^	267 (5%) *^&#^	112 (11%) *^#$^	224 (9%) *^#$^
Obesity	355 (18%)	672 (5%) *	721 (5%) *	110 (2%) *^&#^	41 (4%) *^$^	149 (6%) *^$^
Diabetes	301 (15%)	941 (8%) *	606 (5%) *^&^	212 (4%) *^&^	73 (7%) *^#$^	162 (7%) *^#$^
Contact with a suspicious case	833 (62%)	7202 (59%)	5198 (39%) *^&^	2125 (41%) *^&^	236 (24%) *^&#$^	548 (22%) *^&#$^

* *p* < 0.01 vs. wave 1, ^&^ *p* < 0.01 vs. wave 2, ^#^ *p* < 0.01 vs. wave 3, ^$^ *p* < 0.01 vs. wave 4, ^≠^
*p* < 0.01 vs. wave 5.

**Table 3 healthcare-12-00764-t003:** Sociodemographic characteristics and morbidities of the study population (positive and negative test results for each wave). Data are shown as absolute value and percentage or median (25th–75th percentile).

	Wave 1		Wave 2		Wave 3		Wave 4		Wave 5		Wave 6	
**Variable**	**Positive** (N = 1119)	**Negative** (N = 874)	**Positive** (N = 6372)	**Negative** (N = 5363)	**Positive** (N = 5435)	**Negative** (N = 7658)	**Positive** (N = 1150)	**Negative** (N = 3229)	**Positive** (N = 298)	**Negative** (N = 616)	**Positive** (N = 960)	**Negative** (N = 1532)
Age (years)	48 (37–60)	43 * (34–55)	40 (30–52)	39 * (28–51)	34 (25–46)	36 * (26–49)	39 (29–51)	38 (29–50)	49 (33–66)	40 * (29–55)	45 (33–58)	40 * (30–55)
Sex		*				*						
Female	45.0%	56.4%	54.2%	53.8%	52.5%	56.8%	53.0%	53.7%	57.4%	59.3%	60.0%	59.7%
Male	55.0%	43.6%	45.8%	46.2%	47.5%	43.2%	47.0%	46.3%	42.6%	40.7%	40.0%	40.3%
Smoker	11.9%	12.2%	11.2%	8.5% *	8.4%	7.4%	11.6%	11.0%	7.1%	2.3% *	5.8%	6.3%
Hypertension	21.6%	14.4% *	10.3%	7.5% *	6.5%	5.8%	7.0%	4.9% *	20.3%	5.5% *	9.4%	8.8%
Obesity	19.0%	16.1%	7.1%	3.1% *	6.6%	4.6% *	3.6%	2.0% *	7.8%	2.3% *	5.8%	6.1%
Diabetes	20.0%	8.9% *	8.6%	6.3% *	5.3%	3.9% *	5.3%	3.9%	11.9%	3.9% *	5.8%	7.0%
Contact with a suspicious case	58.0%	65.0% *	54.7%	62.7% *	42.5%	37.1% *	44.3%	37.1% *	37.5%	18.5% *	25.6%	20.5% *

* *p* < 0.01 compared with the positive of the same wave.

**Table 4 healthcare-12-00764-t004:** Summary of the performance of the SARS-CoV-2 individual test prediction models.

	Wave 1	Wave 2	Wave 3	Wave 4	Wave 5	Wave 6
R^2^	0.172	0.267	0.405	0.201	0.535	0.051
Overall prediction (%)	65.7%	71.1%	77.0%	76.6%	82.4%	62.7%
Sensitivity	56.0%	71.0%	63.7%	34.1%	71.4%	18.2%
Specificity	73.9%	71.1%	86.2%	91.7%	87.8%	91.0%
Positive predictive value	64.8%	74.4%	76.1%	59.3%	74.3%	56.2%
Negative predictive value	66.3%	67.5%	77.4%	79.6%	86.1%	63.6%

## Data Availability

The database used in this study is publicly available on the website of Mexico’s Epidemiological Surveillance System [3].

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
