# Peer review of "Individualized Prediction of SARS-CoV-2 Infection in Mexico City Municipality during the First Six Waves of the Pandemic"

_healthcare, 2024, doi:10.3390/healthcare12070764_

Round 1

Reviewer 1 Report

Comments and Suggestions for Authors

To: Healthcare MDPI

Dear EIC,

Dear AE,

This is my review results for the manuscript ID: healthcare-2894212

This study assayed the prediction power of COVID-19-associated diagnostic systems through multivariate binary logistic regression models. The methodology was checked and approved. The manuscript flow and overall writing style were good. Tables and figures sufficiently covered the findings of the study. This study can add some novel things to the field. However, I find some minor comments described below.

Comments

·      The introduction section should be truncated to less than one page.

·      Please remove “non-experimental" from section 2.1.

·      Please remove “analytical 102 (statistical models through binary logistic regression)” from section 2.1 (this should be described in section 2.3.).

·      Please describe the full name of “SISVER/SINAVE” in the first use in the manuscript body (section 2.2); however, described in the abstract section.

·      The conclusion section can be improved by suggesting more prediction tools for COVID-19. Also, recommendations should not be neglected.

Author Response

Comment 1: This is my review results for the manuscript ID: healthcare-2894212

This study assayed the prediction power of COVID-19-associated diagnostic systems through multivariate binary logistic regression models. The methodology was checked and approved. The manuscript flow and overall writing style were good. Tables and figures sufficiently covered the findings of the study. This study can add some novel things to the field. However, I find some minor comments described below.

Comments ·The introduction section should be truncated to less than one page.

Response: We appreciate the Reviewer's remarks, which helped improve our manuscript's presentation and clarity. The Introduction section was shortened to a more concise version, as suggested (pages 1 and 2).

Comment 2: ·      Please remove “non-experimental" from section 2.1.

Response: The word “non-experimental was removed (line 105).

Comment 3: ·      Please remove “analytical 102 (statistical models through binary logistic regression)” from section 2.1 (this should be described in section 2.3.).

Response: The change was applied as suggested (lines 105 to 106).

Comment 4: ·      Please describe the full name of “SISVER/SINAVE” in the first use in the manuscript body (section 2.2); however, described in the abstract section.

Response: The SISVER/SINAVE full name is now described on lines 117 to 118.

Comment 5: ·      The conclusion section can be improved by suggesting more prediction tools for COVID-19. Also, recommendations should not be neglected.

Response: We suggested structural equation modeling as other prediction tools (line 417). We also added recommendations (lines 418 to 427): “Moreover, our findings show high heterogeneity in the variables that comprised each model of the six waves, highlighting the dynamic change in the primary factors driving the epidemic. Therefore, we recommend that further research on prediction models of SARS-CoV2 infection should consider from the start all the symptoms reported in the SINAVE database and other factors such as age, sex, and comorbidities. Our results encourage the future inclusion in the SINAVE database of information regarding the vaccination history and the severity of the symptoms. This may be relevant for models to predict the acute infection with SARS-Cov2 (as in the present work) and for predicting long-term outcomes (such as long-term COVID syndrome) based on the early response during the acute infection.”

Reviewer 2 Report

Comments and Suggestions for Authors

Thanks for the submission. I have some comments for you:

1. The idea you worked on is pretty okay. However, I see no practical applicability with such few performances.

2. I am okay with the data collection way. But the Method you showed is too weak.

3. You wrote, "The objective of this investigation was to develop models for individualized prediction of SARS-CoV-2 infection in a large municipality of Mexico." now, I hardly see any novel method. If you notice carefully, you explained your Method in 10 or 12 lines, which is also holistic.

4. Besides the dataset, this study has no novelty. It's a simple statistical analysis that is also precise.

5. The major fault lies in Method. Other sections are fine.

Comments on the Quality of English Language

Need to check for minor errors.

Author Response

Comment 1: Thanks for the submission. I have some comments for you:

The idea you worked on is pretty okay. However, I see no practical applicability with such few performances.

Response: We appreciate all the comments from the Reviewer, which helped to improve our manuscript. As we already mentioned in the manuscript, the approach proposed here has the potential to be used to provide early diagnosis, which could be useful to improve the care of potentially infected patients to avoid further virus propagation (lines 281 to 282). Our study proposes a COVID-19 detection model that could be an accessible and cost-effective alternative to PCR tests, which is most relevant for vulnerable populations with difficult access to clinical facilities (either public or private). We discuss this now with more detail in lines within a subsection called “4.3 Potential practical application” (lines 360 to 384).

Comment 2: I am okay with the data collection way. But the Method you showed is too weak.

Response: We added a new paragraph in the Discussion section (lines 308 to 318) to specify how starting from the most straightforward statistical approach (binary logistic multivariate regression models), we can identify the variables that predict the individual test and how these models change at the different waves.

Our work was not oriented on a methodological novelty but rather on using simple enough but precise methods to produce valuable insights about the better predictors of SARS-CoV-2 infection in a relevant urban population. For clarity, a subsection is now defined as “4.2. Comparison to other models for personalized prediction of SARS-CoV-2 infection” (lines 286 to 318), where our results are contrasted to previous works with different methods, including advanced ones based on artificial intelligence techniques.

Comment 3: You wrote, "The objective of this investigation was to develop models for individualized prediction of SARS-CoV-2 infection in a large municipality of Mexico." now, I hardly see any novel method. If you notice carefully, you explained your Method in 10 or 12 lines, which is also holistic.

Response: We understand that the Reviewer refers to the description of the binary logistic regression analyses we performed (lines 162 to 168), and by holistic, the Reviewer probably meant that this method produces models to explain the SARS-CoV-2 test result based on data from a population that can explain the variance on the test results with a certain degree (R2). Although, from that point of view, the method is holistic, our work focuses on the potential of the models to predict individual tests, which may be helpful tools for early diagnosis that could prevent the further spread of contagions by opportune isolation measures in each diagnosed patient. 

Comment 4: Besides the dataset, this study has no novelty. It's a simple statistical analysis that is also precise. The major fault lies in Method. Other sections are fine.

Response: As described in the response to Comment 2 above, our work was not oriented to methodological novelty but on obtaining new insights about the variables that better predicted individual infections along the first six waves of the pandemic in a relevant urban population. As mentioned in the discussion section's first paragraph, which is now defined as subsection “4.1. Main contribution”:

“The prediction models showed very good performance, so they are a potential new tool to deal with new waves of contagion. They promote an early risk diagnosis and thus favor measures that reduce infections. The models also show that the prediction variables varied widely between the six waves, highlighting the continuous change in the factors that influence the spread of COVID-19, and hence the importance of considering statistical models for individual prediction that can be updated as these factors evolve with data from the same population.”

Comment 6: Comments on the Quality of English Language: Need to check for minor errors.

Response: The manuscript was reviewed entirely to identify and correct minor errors in the English language.

Reviewer 3 Report

Comments and Suggestions for Authors

In the manuscript entitle “Individualized prediction of SARS-CoV-2 infection in Mexico City Municipality during the first six waves of the pandemic” the authors have developed models for prediction of COVID-19 in a large municipality of Mexico. The authors developed the models for six different pandemic waves that occurred between 2020 and 2023 and observed that there was a large variation in prediction variables among these models indicating the continuous changes in the factors that are involved in COVID-19 infections. Although the test prediction models showed a good percentage of global prediction, the models are lacking some critical and important variables which is important to include as this is the first time comparing the disease models among 6 waves separately. Specific points are as follows:

Major comments:

1.     Vaccination status is one of the strongest variables in predicting the disease and infection.  This manuscript is comparing the models representing different waves between 2020-2023. As vaccination status is a primary factor that have impacted the infection and disease outcomes between these waves, thus it would be difficult to make proper conclusion and comparison without including vaccination information in these models.

2.     The models are lacking some of the important symptoms and variable that predict the COVID19 infection such as lethargy or extreme tiredness, loss of taste, hospitalization, syncope and Delirium or brain fog (especially in elderly patients). Including these might increase sensitivity and specificity.

Minor comments:

1.     Line 311, it should be “6 waves separately” instead of “5 waves separately.”

Comments on the Quality of English Language

I do not have anything to comment on English.

Author Response

Comment 1: In the manuscript entitle “Individualized prediction of SARS-CoV-2 infection in Mexico City Municipality during the first six waves of the pandemic” the authors have developed models for prediction of COVID-19 in a large municipality of Mexico. The authors developed the models for six different pandemic waves that occurred between 2020 and 2023 and observed that there was a large variation in prediction variables among these models indicating the continuous changes in the factors that are involved in COVID-19 infections. Although the test prediction models showed a good percentage of global prediction, the models are lacking some critical and important variables which is important to include as this is the first time comparing the disease models among 6 waves separately.

Specific points are as follows:

Major comments:

  1. Vaccination status is one of the strongest variables in predicting the disease and infection. This manuscript is comparing the models representing different waves between 2020-2023. As vaccination status is a primary factor that have impacted the infection and disease outcomes between these waves, thus it would be difficult to make proper conclusion and comparison without including vaccination information in these models.

Response: Thank you for this comment. We agree with the reviewer about the importance of vaccination as a primary factor impacting the disease and infection. We recognize that the lack of information on vaccination is a significant limitation of our study (lines 398 to 400). Vaccination-related information was not considered in the present work, as it was included in the SINAVE databases only recently, and no data were available for most waves. We expect that our results will encourage including information regarding vaccination in the SINAVE database as an essential measure that could be useful in future research to improve the prediction models, as mentioned now in lines 422 to 424.

Comment 2: The models are lacking some of the important symptoms and variable that predict the COVID19 infection such as lethargy or extreme tiredness, loss of taste, hospitalization, syncope and Delirium or brain fog (especially in elderly patients). Including these might increase sensitivity and specificity.

Response: We now mention in lines 403 to 410 the following:

“Finally, the SINAVE database includes information on the symptoms' presence but not the severity, which may help to improve the prediction accuracy. However, our study was primarily based on patients from outpatient clinics, where the severity of symptoms is mild or moderate and comprise the most significant proportion of infected patients in the overall population. Further studies are required to test the feasibility of using the presence of early symptoms during the acute infection to predict severe chronic symptoms, such as fatigue, syncope, and delirium or brain fog, associated with the development of long-COVID syndrome.”

Comment 3: Minor comments:

  1. Line 311, it should be “6 waves separately” instead of “5 waves separately.”

Response: The line was corrected (line 334).

Round 2

Reviewer 2 Report

Comments and Suggestions for Authors

Dear authors, I have some comments:

1. First, I do not expect some extraordinary work. But I expect decent work of a certain level.

2. Your method is weak due to several reasons:

i) It's too general and doesn't have much testing.

ii) There is model selection bias.

iii) A p-value threshold of < 0.01 is fine, but you must be careful. Several waves may vary. Also, there should be some comparative analysis.

iv) Why are there six different models than 1 model for all waves? What you want to do, and Hosmer-Lemeshow is okay but not in sufficient condition alone. Influential outliers and residual analysis may not be covered using Hosmer-Lemeshow.

v) 95% CIs is standard practice. In particular, there may be 10 cases of Covid-19 in 10,000 people. The model needs to be sensitive.

vi) I can apply the deep learning model and produce better results. But you haven't tried it.

vii) There needs to be I just wanted to let you know a clear view of the method in this work. 

I just wanted to let you know that I am okay with you saying the method is not your primary focus. However, the method should be well-tested, compared, and free from biases. Even though you say the method is not the central pillar, it is. The success of prediction is critical in infectious diseases. A wrong prediction means one person infects 20 more, and so on. Also, whatever method you use, you need to defend the usage of things like Hosmer-Lemeshow, why Logistic Regression, conditional forward method, etc. I always need evidence or literature on these.

Comments on the Quality of English Language

Proofread needed.

Author Response

Comment 1: Dear authors, I have some comments:

  1. First, I do not expect some extraordinary work. But I expect decent work of a certain level.
  2. Your method is weak due to several reasons:
  3. i) It's too general and doesn't have much testing.
  4. ii) There is model selection bias.

Response: Additional testing of the models was added in the manuscript, as described below.

Selection bias refers to selecting the variables from a data set. As a first approximation, we chose the most common approach of identifying the variables that significantly differed between patients with a positive test result and patients with a negative result from the entire data set of each wave. These variables were the potential independent variables introduced in the binary regression models, using the forward conditional method to test the effect of each added variable on the model’s goodness of fit. We realize that more robust approaches, such as cross-validation or bootstrapping, are needed to address potential model selection bias. However, this is not included in the present work to avoid an increased complexity of this first approximation. Still, it is now mentioned in section 4.3, “Study limitations,” as future work that is needed (lines 436 to 443).

Comment 2: iii) A p-value threshold of < 0.01 is fine, but you must be careful. Several waves may vary. Also, there should be some comparative analysis.

Response: Different models were adjusted because the waves could vary. For comparative analysis, we verified whether the confidence intervals for the odds ratio in the different waves overlap (Supplementary material, Table S2). The odds ratio of age during wave 5 was larger than during wave 3. The odds ratio of being a smoker was lower during wave 3 than during wave 2. Compared to waves 1 and 2, the odds ratio of cough was consistently higher during waves 3, 4, and 5. Odynophagia also had a higher odds ratio during waves 3 and 4 than during wave 2. The headache had higher odds ratios during waves 2, 3, and 5 than during wave 1. The odds ratio of fever during wave 3 was larger than in Wave 2, and Wave 5 also had a larger odds ratio of fever than Wave 2 and Wave 4. Finally, the odds of irritability were larger in wave 4 compared to waves 2 and 3. This is now described in lines 264 to 272 of the Result section. These comparisons are now discussed in lines 382 to 392.

Comment 3: iv) Why are there six different models than 1 model for all waves? What you want to do, and Hosmer-Lemeshow is okay but not in sufficient condition alone.

Influential outliers and residual analysis may not be covered using Hosmer-Lemeshow.

Response: We agree with the Reviewer regarding the unnecessary adjustment of a general model for all waves, considering that we obtained a different model for each wave and the effect of a larger sample size in the general model. Therefore, we have eliminated all comments regarding a “general model” from the manuscript.

Although having influential outliers in our models is unlikely because almost all variables are categorical, the Cook statistic for influence was calculated for the final model of each wave, and we did not find any influential observations (Supplementary Material, Figure S1). This is now mentioned in lines 285 to 287.

Comment 4: v) 95% CIs is standard practice. In particular, there may be 10 cases of Covid-19 in 10,000 people. The model needs to be sensitive.

Response: We agree that increasing the sensitivity of the prediction is essential. The present work is the first stem, based on a straightforward statistical approach, and testing with more advanced methods to improve the accuracy is an important direction for future studies, as mentioned in lines 327 to 337. 

Comment 5: vi) I can apply the deep learning model and produce better results. But you haven't tried it.

Response: We decided to use logistic regression because of its broader knowledge, and we think it is a good approximation for what we wanted to do.

Comment 6: vii) There needs to be I just wanted to let you know a clear view of the method in this work.

Response: We have improved the description of the methods in the revised manuscript.

Comment 7: I just wanted to let you know that I am okay with you saying the method is not your primary focus. However, the method should be well-tested, compared, and free from biases. Even though you say the method is not the central pillar, it is. The success of prediction is critical in infectious diseases. A wrong prediction means one person infects 20 more, and so on. Also, whatever method you use, you need to defend the usage of things like Hosmer-Lemeshow, why Logistic Regression, conditional forward method, etc. I always need evidence or literature on these.

Response: We agree with the Reviewer on the importance of aiming for better prediction methods. In the revised manuscript, we incorporated more model testing (such as the test for influential points). When additional testing was not feasible for the current work, we noted the need for these in future work, as explained above.

Comment 8: Comments on the Quality of English Language: Proofread needed.

Response: The manuscript was proofread in English.

Reviewer 3 Report

Comments and Suggestions for Authors

The authors have addressed my previous concerns in their response and included in the text.

Author Response

Comment 1: The authors have addressed my previous concerns in their response and included in the text.

Response: Thank you.